# A Two-Stage Rolling Bearing Weak Fault Feature Extraction Method Combining Adaptive Morphological Filter with Frequency Band Selection Strategy

**Jun Li [1], Hongchao Wang [2,*], Simin Li [3], Liang Chen [3] and Qiqian Dang [3]**

[1] School of Mechatronic Engineering, China University of Mining and Technology, 1 Daxue Road, Xuzhou 221116, China
[2] Mechanical and Electrical Engineering Institute, Zhengzhou University of Light Industry, 5 Dongfeng Road, Zhengzhou 450002, China
[3] Zhengzhou Research Institute of Mechanical Engineering Co., Ltd., 149 Science Avenue, Zhengzhou 450001, China
[*] Correspondence: hongchao1983@126.com

**Abstract:** To extract the weak fault features hidden in strong background interference in the event of the early failure of rolling bearings, a two-stage based method is proposed. The broadband noise elimination ability of an adaptive morphological filter (AMF) and the superior capability of a frequency band selection (FBS) strategy for fault transient location identification are comprehensively utilized by the proposed method. Firstly, the AMF with a simple theory and high calculation efficiency is used as a preprocessing program to enhance the fault transient features. Then, the proposed FBS strategy based on the sparsity index (SI) is utilized to further handle the filtered signal processed by the AMF. Finally, the constructed optimum bandpass filter based on the analysis result of the FBS is used to further filter the handled signal processed by AMF and envelope spectral analysis is applied on the last filtered signal to realize the ideal fault feature extraction effect. Compared with the other traditional FBS methods based on kurtosis or the other index, the proposed FBS strategy based on SI has strong robustness to noise. One experimental signal and one engineering vibration signal are used, respectively, to verify the feasibility of the proposed method.

**Keywords:** weak fault; adaptive morphological filter; feature extraction; frequency band selection; rolling bearing



## 1. Introduction

Envelope spectral (ES) analysis is one kind of classical and effective fault feature extraction method for rolling element bearing. However, its effect will be weakened under the influence of strong interferences. Usually, the fault information sensitive frequency band is needed to be selected out of the full band firstly and then ES analysis is applied on the fault information sensitive frequency band to receive better feature extraction results. The above process is essentially named as FBS. The typical signal characteristic of a faulty rolling bearing is impulse or transient, so kinds of impulse sensitive indexes are used for FBS. Spectral kurtosis (SK), as proposed by Antoni [1,2] with its fast implementation algorithm, that is kurtogram [3], is the pioneer of FBS; it uses the temporal kurtosis to reflect the amount of transient in each divided frequency band. In some conditions such as strong background noise containing random impulse peaks, invalid results will be yielded by SK. To overcome the above shortcomings of SK, ES analysis is applied on the faulty signal firstly and the kurtosis of the envelope result is calculated subsequently for FBS; the Protrugram [4] was proposed, which had superior performance to detect impulses with a low ratio of signal-to-noise. In [5], the concept of the negentropy index of the faulty bearing signal is introduced and used as an index for FBS, which could reflect the impulse

and cyclostationarity of a bearing fault signal simultaneously. Besides, the concept of infogram based on the introduced negentropy index was also introduced, which extended the applicability domain of traditional SK. Besides, kinds of related improved methods based on kurtogram such as the fast kurtogram by introducing genetic algorithm [6], the adaptive SK [7], the improved kurtogram adopting wavelet packet transform as filter [8], the improved kurtogram adopting robust local mean decomposition as filter [9], and so on have also been arising.

Inspired by the above FBS methods, a new FBS method based on SI is introduced in this study due to the reason that SI has stronger resistance to noise and random shock than the other impulse sensitive indexes such as kurtosis and negative entropy. However, it should be noted that the effect of FBS would be not ideal when the inferences are too strong, especially in the early weak failure stage of rolling bearing, and the reason is mainly due to that the energy of impulse components is much weaker compared with the other kinds of components, so it is better to preprocess the original vibration signal firstly. Different signal processing methods such as wavelet transform (WT) [10] and empirical mode decomposition (EMD) [11] could be selected for this target. However, as for WT, the optimal wavelet basis is needed to be selected in advance, which is impossible in most engineering cases because the prior knowledge of the diagnosis object is unknown. EMD has the following two fatal flaws: (1) its mathematical theoretical support is relatively weak [12]; (2) it is sensitive to noise [13]. Although the above two defects of EMD have been solved to some extent by the subsequently emerging improved EMD methods such as EEMD [14], CEEMDAN [15], and so on, their huge and complex calculations limit their further engineering application. In view of the diversity characteristics of the target signal, the analysis dictionary based on sparse representation methods [16–18] could use a series of different wavelet bases to match the characteristics of the target signal and the above drawbacks of WT and EMD are solved to some extent. Unfortunately, the multi-wavelet basis functions still need to be constructed artificially. Though the multi-wavelet basis functions could be learned adaptively according to the analyzed signal itself by the self-learning dictionary based sparse representation methods [19–21], they have the defect of requiring a huge amount of calculation. In recent years, as a new emerging signal processing method, variational mode decomposition (VMD) [22,23] has been used widely in denoising vibration signals of rotating machinery. However, as the two key parameters of VMD, the penalty factor and the number of modes should be selected appropriately, which inferences the denoising result of VMD directly [13]. Morphological filtering (MF) is an important nonlinear filtering method based on mathematical morphological transformation [24]. MF uses the same structure elements (SEs) as the filtering window to match the signal to be analyzed or modify the signal locally in the time domain [25] and the information is preserved when the local morphological features of the analyzed signal match the SEs. Its working principle is simple and the calculation theory is efficient. Initially, MF was used widely in other signal processing fields such as the image signal and power signal [26]. MF could extract the edge contour and the shape features of the vibration signal effectively and realize the retention of fault features and the removal of interference noise. Recently, various examples in the literature have been emerging on the application of MF in mechanical fault diagnosis and various satisfactory results have been achieved. However, the filtering effect of MF in the fault diagnosis area is mainly determined by the selection of the scale and the shape of the SE of MF; different kinds of corresponding research have been carried out. An empirical rule was provided in [27] for choosing the shape and scale of SE. Filter the original signal in different scales and their corresponding weighted average is calculated to alleviate the scale selection influence of SE [28,29] and its defects include that its process is time-consuming and fault-unrelated components might be contained [30]. Though some research [31–34] reducing the fault-unrelated components problem existed in [28,29], the problem of low computation effect was still not resolved.

To enhance the adaptive ability of MF in the feature extraction of rolling bearing, an AMF with simple theory and high calculation efficiency is used as pre-processing step of

the proposed FBS in the paper. The main contributions of the paper are as follows: (1) The impulse characteristic components buried in the original multi-component vibration signal of faulty bearing is enhanced by the AMF algorithm adaptively. (2) SI is introduced in the FBS algorithm, which has more robustness to noise and random impulses. (3) The feature extraction double enhancement method for the rolling bearing's weak fault by combining the AMF and FBS is proposed in the paper, whose feasibility and effectiveness are verified through one experiment vibration signal and one engineering vibration signal, respectively.

The organizations of the remaining paper is as follows: the calculation processes of AMF and FBS are presented in Sections 2 and 3, respectively. The overall flow chart of the proposed method and its details are elaborated in Section 4. The effectiveness of the proposed method is verified through an experimental signal and an engineering signal in Section 5. The comparison study is carried out in Section 6 and the conclusion is obtained in Section 7 at last.

## 2. AMF

### 2.1. Basic Theory of MF

There are four kinds of basic operators in MF: dilation, erosion, opening, and closing. The one-dimensional original signal is denoted by $f(n)$, whose discrete form could be defined as $F = (0, 1, \cdots, N-1)$. $g(m)$ is also a one-dimensional signal, whose discrete form is defined as $G = (0, 1, \cdots, M-1)(N \geq M)$. $g(m)$ is the SE and the above four basic operations of mathematical morphology are defined as follows:

Dilation:

$$(f \oplus g)(n) = \max\{f(n-m) + g(m)\}\{1 \leq n \leq N; 1 \leq m \leq M\} \tag{1}$$

Erosion:

$$(f\Theta g)(n) = \min\{f(n+m) - g(m)\}\{1 \leq n \leq N; 1 \leq m \leq M\} \tag{2}$$

Opening:

$$(f \circ g)(n) = (f\Theta g \oplus g)(n) \tag{3}$$

Closing:

$$(f \bullet g)(n) = (f \oplus g\Theta g)(n) \tag{4}$$

where $\oplus$, $\Theta$, $\circ$, and $\bullet$ represent the dilation operator, erosion operator, opening operator, and closing operator, respectively.

The following several commonly used morphological operators could be constructed through the cascade combination of the above four morphological operators:

Dilation and erosion gradient operator (GDE) [31]:

$$GDE(n) = (f \oplus g)(n) - (f\Theta g)(n) \tag{5}$$

Closing and opening gradient operator (GCO) [35]:

$$GCO(n) = (f \bullet g)(n) - (f \circ g)(n) \tag{6}$$

Closing–opening and opening–closing gradient operator (GCOOC) [36]:

$$GCOOC(n) = CO(n) - OC(n) \tag{7}$$

Dilation and erosion average-hat operator (AHDE) [37]:

$$AHDE(n) = f(n) - \frac{(f \oplus g)(n) + (f\Theta g)(n)}{2} \tag{8}$$

Closing and opening average-hat operator (AHCO) [38]:

$$AHCO(n) = f(n) - \frac{(f \bullet g)(n) + (f \circ g)(n)}{2} \tag{9}$$

Closing–opening and opening–closing average-hat operator (AHCOOC) [39]:

$$AHCOOC(n) = f(n) - \frac{CO(n) + OC(n)}{2} \tag{10}$$

Although the feature extraction ability of the above kinds of combined morphological operators shown from Equations (5)–(10) could be enhanced greatly, their transient feature extraction performance is still unsatisfactory in the case of strong interference [40]. To solve the above problem, the morphology gradient product operator (MGPO) [40] as in the following is constructed by using the product of GCO and GCOOC:

$$MGPO(n) = GCO(n) \bullet GCOOC(n) \tag{11}$$

The same as the construction ideology of *MGPO*, the *MHPO* [41] is introduced to extract the transient features by integrating AHCO and AHCOOC:

$$MHPO(n) = AHCO(n) \bullet AHCOOC(n) \tag{12}$$

The excellent performance of MGPO and MHPO in transient features extraction over the other traditional morphological operations has been proved and the other two new morphological operations named as MHPO1 and MHPO2 originating from AHDE [37] are introduced for further enhancing the extraction ability of transient features:

$$MHPO1(n) = AHDE(n) \bullet AHCO(n) \tag{13}$$

$$MHPO2(n) = AHDE(n) \bullet AHCOOC(n) \tag{14}$$

In this paper, the MHPO1 shown in Equation (13) is used as a morphological operator for the reason that AHDE and AHCO had been verified much more efficiently than AHCOOC in extracting transient features [37].

### 2.2. Strategy for Designing SE

The triangular SE, semi-circle SE, and flat SE [28] constructed from window functions are commonly used in traditional MFs for vibration signal processing. However, the appropriate shapes and proper length and height parameters need to be selected for a better extraction effect in the above SEs. Unfortunately, it is impossible to gain the prior knowledge of the diagnosed object in most engineering cases. The ratio between the sampling frequency and characteristic frequency is used to determine the information adaptively by some studies [36,39,40], whereas most of them has low computational efficiency, which is not suitable for online application.

Considering the calculation efficiency and determining the information of SE adaptively, a new SE design strategy for MF is used [40], whose basic ideology is shown in Figure 1, and it relies entirely on the intrinsic properties of the analyzed signal. Its concrete steps are as follows:

Step 1: All the local maximum and minimum points of the original collected vibration signal marked as a blue line in Figure 1 are found.

Step 2: The found local maximum and minimum points in step 1 are fitted by using the cubic spline interpolation (CSI) method and the new fitted curve shown as the red line in Figure 1 is obtained.

Step 3: Determine the shape of SE according to the narrow waveform ($a - b - c, c - d - e, e - f - g$, and so on in Figure 1) between the two adjacent minimum points in the fitted curve. Then, the minimum amplitude of each narrow waveform is adjusted to zero. At

last, the adjusted narrow waveforms are measured as SEs to extract the impact components buried in the original vibration signal.

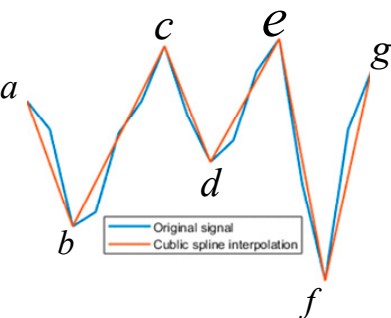

**Figure 1.** Illustration of designing SEs of MF.

A simulated signal $s(t)$ is used to verify the de-noising ability of AMF in the section. The mathematical expressions of the simulated signal $s(t)$ and its four components are provided in Equation (15) and their corresponding time domain waveforms are presented in Figure 2a–e, respectively. In Equation (15), $p(t)$ is the impulse signal and its demodulated amplitude is represented by $A(t)$. $s(t)$ simulates a rolling bearing with a failed inner race and the fault characteristic frequency (FCF) of the inner ring is set as 97 Hz. Theoretically, there exists a random slip between the rolling elements and the raceways and the random slip is represented by $\tau_i$. In Figure 2e, the impulse features are hidden completely by the strong interferences and the direct ES result of $s(t)$ is shown in Figure 2f, on which only the FCF of the inner race could be identified clearly. The AMF is applied on the original signal as shown in Figure 2e and the filtered signal is presented in Figure 2g. Compare Figure 2g with Figure 2e and the impulse features are enhanced evidently. The kurtosis index of the signal shown in Figure 2g increases about 300% compared with the original signal through calculation. The ES analysis result of the de-noised signal is shown in Figure 2h, based on which not only the FCF on the inner ring could be extracted but also its harmonics are also extracted perfectly.

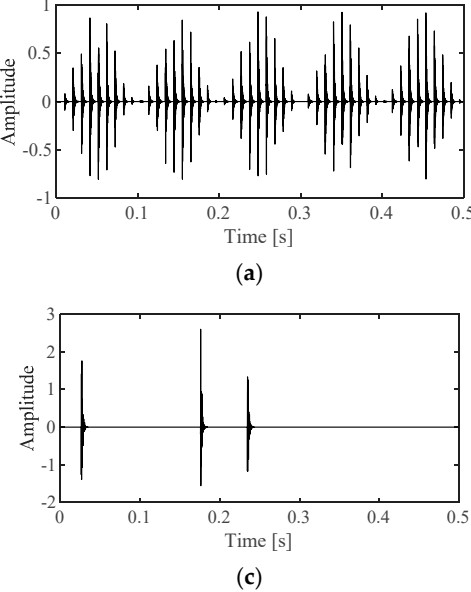

(a)

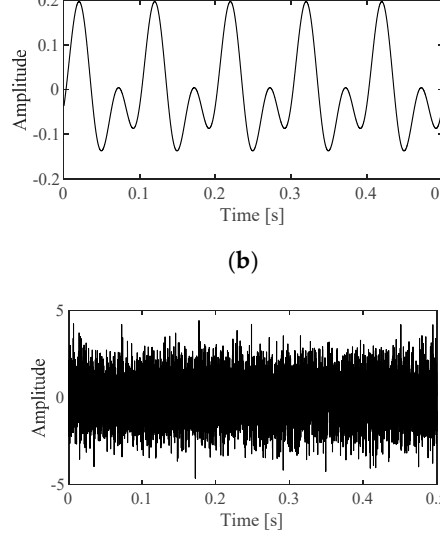

(b)

(c)

(d)

**Figure 2.** *Cont.*

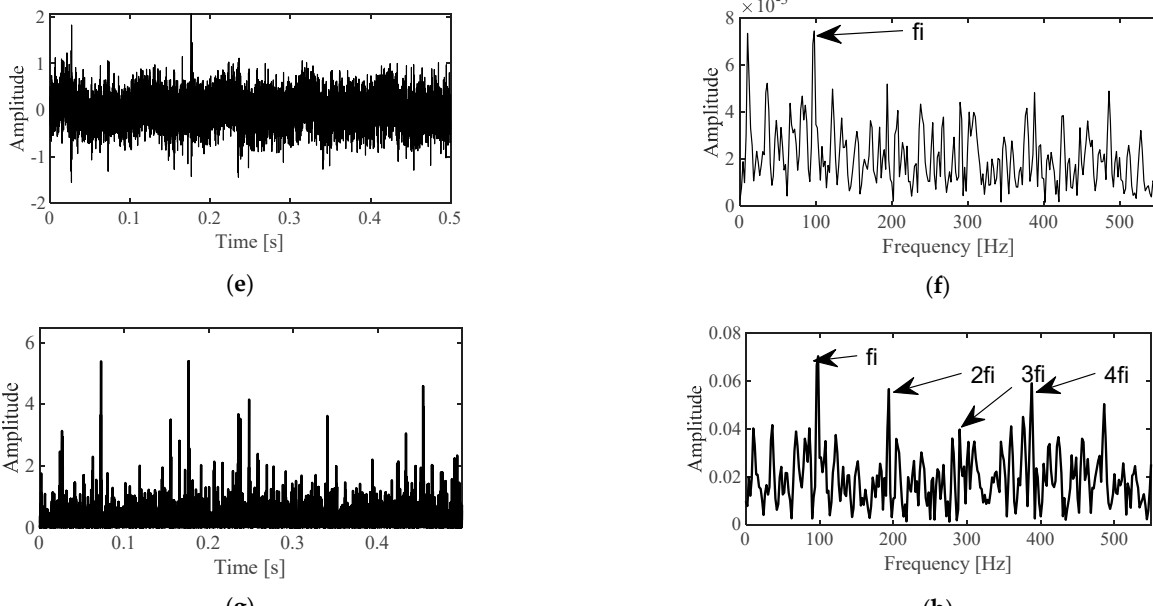

**Figure 2.** The filter result of the rolling bearing simulated signal based the proposed AMF. (**a**) Periodic impulses. (**b**) Harmonic components. (**c**) Random impulses. (**d**) White noise. (**e**) Mixed signal. (**f**) Envelope spectral of the signal shown in (**e**). (**g**) The filtered signal of the mixed signal shown in (**e**) using the proposed AMF. (**h**) Envelope spectral of the signal as shown (**g**).

$$
\begin{cases}
s(t) = p(t) + h(t) + n(t) \\
p(t) = \sum_{i=1}^{97} A(t) \cdot e^{-1000 \cdot (t - i/97 - \tau_i)} \sin[2\pi \cdot 3900 \cdot (t - i/97 - \tau_i)] \\
h(t) = 0.1 \cdot \sin(2\pi \cdot 10 \cdot t + \pi/6) + 0.1 \cdot \sin(2\pi \cdot 20 \cdot t - \pi/3) \\
r(t) = B_r \cdot e^{-800 \cdot (t - T_r)} \sin[2\pi \cdot 2000 \cdot (t - T_r)] \\
A(t) = 0.5 \cdot [1 - \cos(2\pi \cdot 10 \cdot t)]
\end{cases}
\tag{15}
$$

## 3. The FBS Strategy

The proposed FBS strategy mainly consisted of five steps:

Step 1: Segment the AMF filtered signal.

Step 2: Frequency band partition.

Step 3: The SI of each frequency band candidate is calculated.

Step 4: The frequency band candidate with the biggest SI is selected.

Step 5: Apply the ES analysis on the selected fault information sensitive frequency band and the fault features are extracted.

In step 1, supposing the collected original discrete signal is represented as $x[n](n = 1, 2, \cdots, N)$. The signal sampling frequency is noted as $f_s$. The other three parameters are symbolized as follows: $L_{overlap}$ represents the sample overlap length between two neighboring segments; $n_{seg}$ represents the number of divided segments; and $L_{seg}$ represents the length of each segment. A sliding segmentation method, with its mathematical being represented in Equation (16), is used to segment the original signal $x[n]$:

$$
N = (n_{seg} - 1) \times (L_{seg} - L_{overlap}) + L_{seg}
\tag{16}
$$

In step 2, the low-pass/band-pass/high-pass filters are used to filter the original signal to obtain frequency band candidates; $f_c$ in Equation (17) and $\Delta f$ in Equation (18) represent the filters' corresponding center frequency and bandwidth, in which $i$ is the segment index of each band level and $j$ is the filtering structure level and the partition ideology is same as kurtogram [3]. The tree of filter bands as shown in Figure 3 is produced repeatedly in a pyramidal manner.

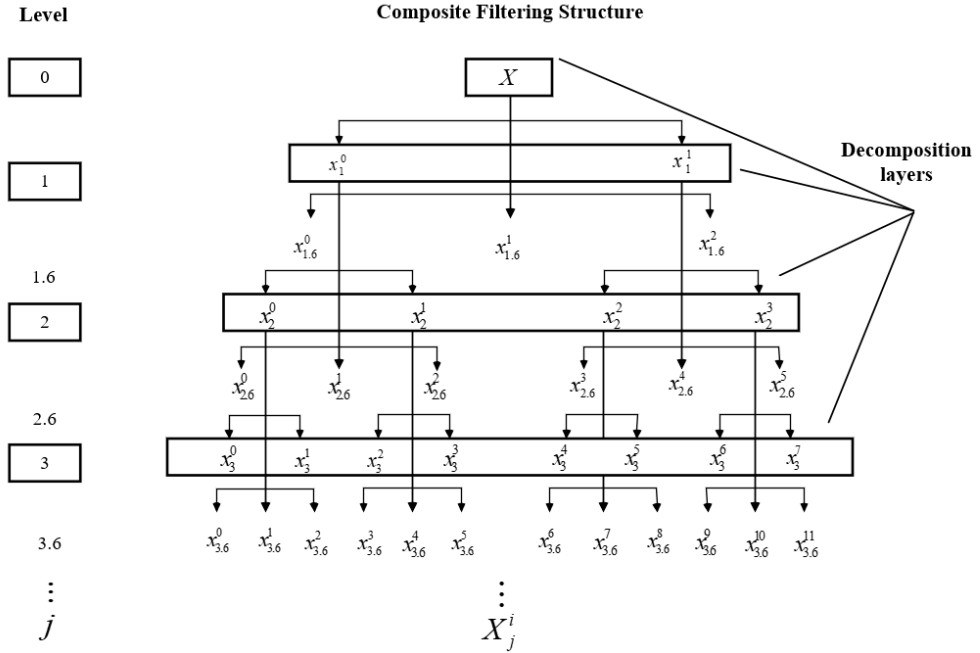

**Figure 3.** The production of three of the filter bands.

$$f_c = \left(i + \frac{1}{2}\right) \times 2^{-j-1} \times f_s \tag{17}$$

$$\Delta f = 2^{-j-1} \times f_s \tag{18}$$

In step 3, the SI, also named GI [42], is used for FBS, whose calculation formula is shown in Equation (19).

$$GI = 1 - 2\sum_{p=1}^{N} \frac{SE^r[p]}{\|SE\|_1}\left(\frac{N - p + 0.5}{N}\right) \tag{19}$$

where $\|\ \|_1$ represents the $l_1$ norm, $SE$ denotes the squared envelope of $x[n]$, and $SE^r$ represents $SE$ in ascending order, that is $SE^r[N] \geq SE^r[N-1] \geq \cdots \geq SE^r[1]$. $SE$ could be calculated by using Equation (20):

$$SE = |\bar{s}|^2 = |s + j \cdot H(s)|^2 \tag{20}$$

where $\bar{s}$ represents the analytical signal of the analyzed signal $s$, $H()$ is the symbol of the Hilbert transform, and $j^2 = -1$.

To verify that the GI has an advantage and higher reliability than the other advanced indexes such as the Hoyer measure [43], L2/L norm [44], and kurtosis index to reflect the cyclic transient features, the five signals as provided in Equation (21) are used and their four indexes are calculated. In Equation (21), Sig1 is the composite signal of two sinusoids, Sig2 is the random white noise, Sig3 represents one impulse hidden in random white noise, Sig4 is two impulses hidden in random white noise, and Sig5 is multiple impulses hidden in random white noise to simulate the periodic shock vibration characteristics of the faulty bearing. The time domain diagrams of Sig1–Sig5 are presented in Figure 4a and their corresponding four indexes are provided in Figure 4b, based on which the phenomenon of the increasing in GI with the much more evident transient features could be observed. However, the other three indexes do not own the virtue of GI.

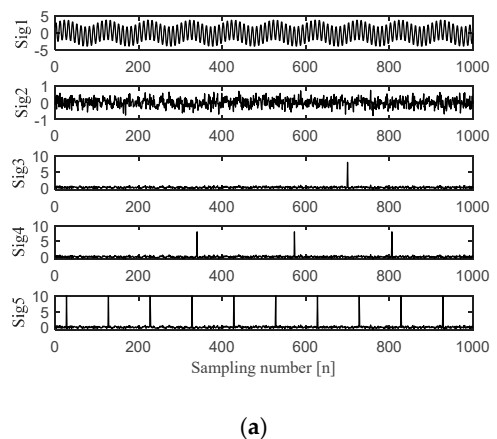

(**a**)

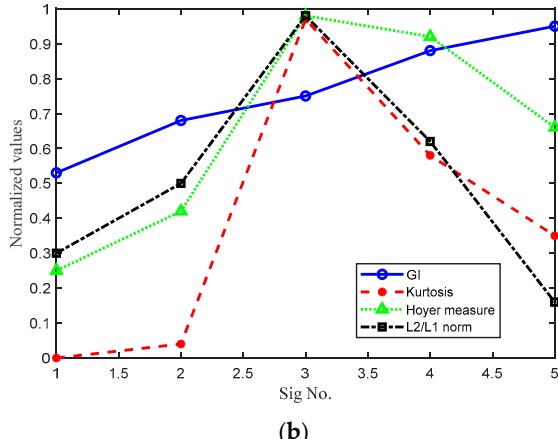

(**b**)

**Figure 4.** Five signals with their four indexes. (**a**) Five signals. (**b**) Four indexes corresponding to the five signals as shown in (**a**).

$$
\begin{cases}
Sig1 = \sin(2\pi/100 * t) + 3\sin(2\pi/10 * t)t = 1 : 1000 \\
Sig2 = 0.25\text{randn}(Sig1) \\
Sig3 = Sig2;\ Sig(700) = 8 \\
Sig4 = Sig2;\ Sig(340 : 233 : end) = 8 \\
Sig5 = Sig2;\ Sig(28 : 100 : end) = 10
\end{cases}
\tag{21}
$$

## 4. Flow Chart of the Proposed Method

To handle the difficult problem of early weak fault feature extraction of rolling bearing, a corresponding two-stage method is proposed by combining the AMF and FBS strategy. The scale and shape of SE could be determined by AMF adaptively based on the intrinsic characteristics of the signal to be analyzed itself, which could reduce the interferences to a maximum extent. To further reduce the influence of noise interference caused by full frequency band ES analysis, the proposed FBS is used to select the fault information's most sensitive frequency band. Finally, ES analysis is implemented on the selected fault information sensitive frequency band and a satisfactory feature extraction result could be obtained. Figure 5 shows the overall flow of the proposed method, whose specific details are as follows:

Step 1: Collect the vibration signal of the faulty rolling bearing using accelerated sensors.

Step 2: The scale and shape of SEs are determined according to the processes in Section 2.2.

Step 3: Apply the MF shown in Equation (13) using the obtained SEs in step 2 on the original vibration signal.

Step 4: Select the fault information sensitive frequency band based on the proposed FBS in Section 3 to further eliminate the noise pollution caused by full frequency band ES analysis.

Step 5: Apply ES analysis on the signal contained in the selected frequency band.

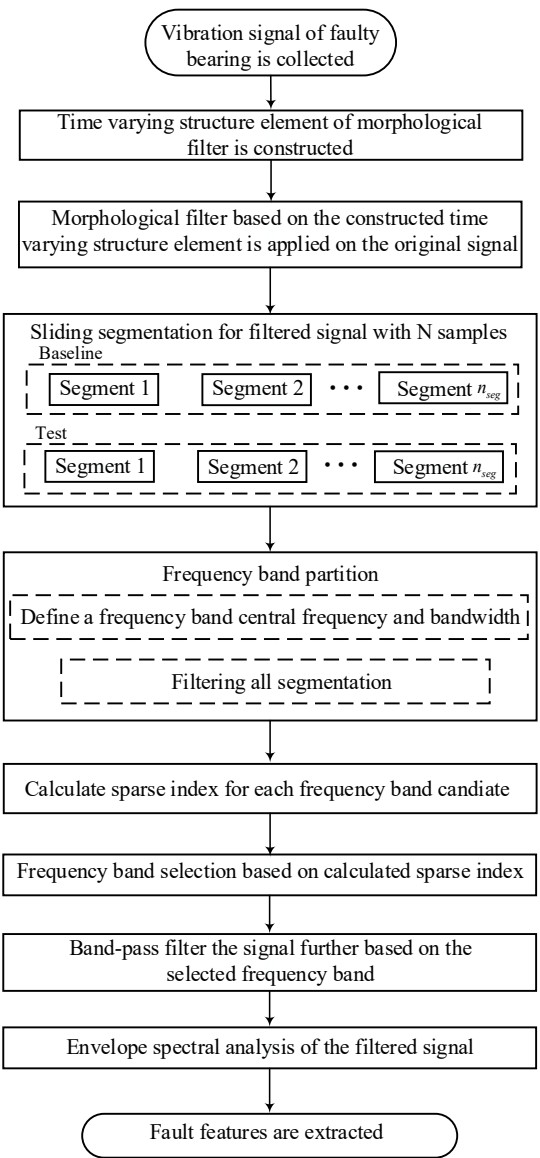

**Figure 5.** Flow chart of the proposed method.

## 5. Verification

### 5.1. Experiment Verification

The rolling bearing life cycle experiment is conducted in the section and the signal collected at the early weak failure stage is analyzed using the combined two-stage method. The accelerated bearing life test (ABLT-1A) is provided by Hangzhou Bearing Test & Research Center. It simultaneously hosts four rolling element bearings on one shaft driven by an AC motor and coupled by rubber belts. The test rig is shown in Figure 6. A new bearing will be installed if one fails. Select one of the same test bearings as [45] as the study object. The parameters and the fault characteristic frequencies of the test rolling bearing are shown in Tables 1 and 2, respectively.

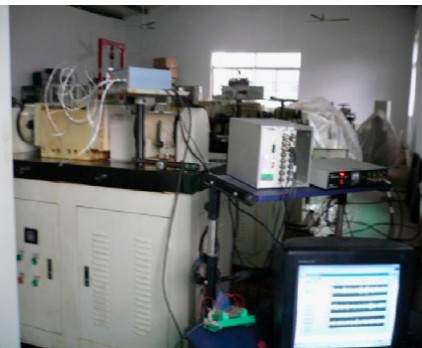

**Figure 6.** The test rig.

**Table 1.** The parameters of the test rolling bearing.

| Type | Ball Number | Ball Diameter (mm) | Pitch Diameter (mm) | Contact Angle | Motor Speed (rpm) | Load (kN) |
|---|---|---|---|---|---|---|
| 6307 | 8 | 13.494 | 58.5 | 0 | 3000 | 12.744 |

**Table 2.** The fault characteristic frequencies of the test rolling bearing.

| $f_r$ | $f_c$ | $f_b$ | $f_i$ | $f_o$ |
|---|---|---|---|---|
| 50 | 19 | 102 | 246 | 153 |

The whole life cycle of the selected test bearing is 2469 min. The vibration data corresponding to the 2297th minute are used as the analyzed object, whose time-domain diagram is presented in Figure 7a. It is found that failure arises on the inner ring of the test bearing by disassembling it after experiment and the FCF of the inner race is 245 Hz. The transient characteristics of the faulty bearing are obscured for the effect of strong interferences and the judgement of the fault location occurring on the inner race would not be realized based on the periodic interval between the two adjacent impacts in the waveform as presented in Figure 7a. Figure 7b is the full-band ES analysis result and only the spectral energy distribution characteristics between 0 Hz and 800 Hz are displayed for convenient analysis. Unfortunately, any useful fault characteristic information could not be obtained based on Figure 7b by applying ES analysis over the full frequency band and the main reason is still due to the strong inferences.

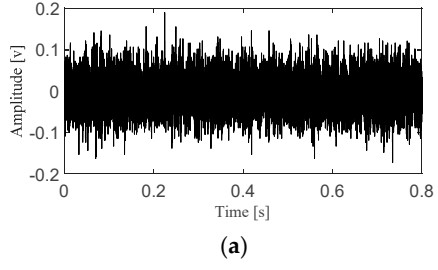

(**a**)

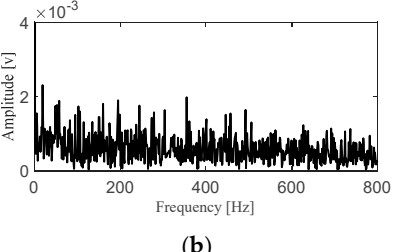

(**b**)

**Figure 7.** *Cont.*

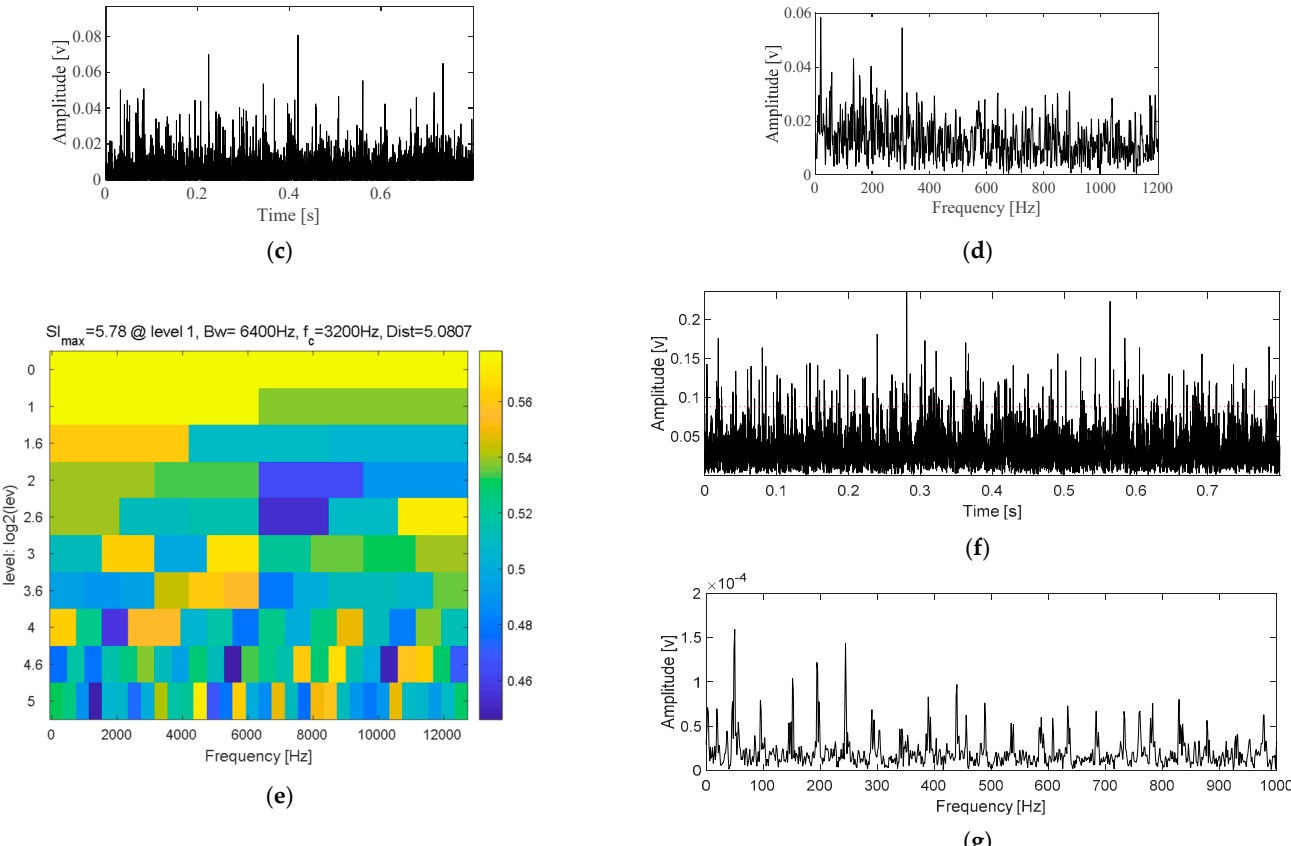

**Figure 7.** Experiment verification of the proposed method. (**a**) The original vibration signal of the experimental bearing at its early weak fault stage. (**b**) Envelope spectral of the signal shown in (**a**). (**c**) The filtered signal of the signal shown in (**a**) using AMF. (**d**) Envelope spectral of the signal shown in (**c**). (**e**) Analysis result of the signal shown in (**c**) using the proposed FBS method. (**f**) Envelope of the filtered signal of the signal shown in (**c**) using the proposed FBS method. (**g**) FFT of the signal shown in (**f**).

Based on the described method in Figure 5, AMF is used to preprocess the original signal initially and the de-noised signal is shown in Figure 7c. By comparing Figure 7c with Figure 7a, the transient characteristics of the former are enhanced significantly and the kurtosis index of the de-noised signal is about 5.2 times that of the original signal through statistical calculation, which proves the enhancement effect of AMF on the transient characteristics of the faulty bearing. To verify the necessity of further processing on the de-noised signal, full-band ES analysis is carried out on the signal as shown in Figure 7c,d, which show the corresponding result: although the FCF of inner race is extracted, its harmonics still could not be identified effectively due to the inference of strong interferences, and further FBS is needed to handle de-noised signal. Apply FBS on the de-noised signal and Figure 7e presents the analysis result, based on which the key parameters (center frequency $f_c$ and bandwidth $B_w$) of the selected optimal frequency band with the biggest SI (SI = 5.78) for ES analysis are as follows: $f_c$ = 3200 Hz and $B_w$ = 6400 Hz. According to the above two obtained parameters, a band-pass filter is constructed to further filter the signal shown in Figure 7c and the envelope result of the further de-noised signal is displayed in Figure 7f. Apply FFT on the envelope signal shown in Figure 7f and the last fault features extraction effect is shown in Figure 7g, based on which both the FCF of the inner race and its harmonics are extracted perfectly.

### 5.2. Engineering Verification

The engineering test object is a water pump unit of a paper mill, whose structure diagram is shown in Figure 8: the centrifugal water pump is driven directly by motor. The rotating speed and rated power of the driving motor are 1480 RPM and 250 kW, respectively. The unit is tested by the off-line signal collection and analysis equipment produced by Zhengzhou Expert Technology Co., Ltd. An acceleration sensor is used and its model and sensitivity are EAG01-100 and 100 mv/g, respectively. The measuring points of the unit and the amplitudes of each measured points are displayed in Table 3. The sampling frequency is set as $f_s$ = 6400 Hz. Based on Table 3, the vibration amplitude corresponding to the driving end of the pump is the largest. The bearing locating on the driving end of the pump is disassembled after shutting off the unit and it is found that the fault arises on the inner ring. The disassembled bearing and its fault location are shown in Figure 9 The type of the bearing and its FCFs are detailed in Table 4. The channel with the largest vibration value at the driving end of the pump in Table 3, that is, the horizontal direction channel is chosen for analyzing.

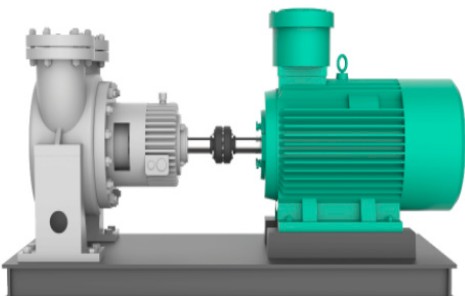

**Figure 8.** The monitored engineering object.

**Table 3.** The collected vibration values of the monitored machine.

| Number | Measured Points | Direction | Measured Values (mm/s$^2$) |
|---|---|---|---|
| 1 | Free end of motor | Horizontal | 13.9 |
| 2 | Drive end of motor | Horizontal | 12.8 |
| 3 | | Vertical | 11.3 |
| 4 | | Axial | 8.0 |
| 5 | Driving end of pump | Horizontal | 45.9 |
| 6 | | Vertical | 37.7 |
| 7 | | Axial | 28.7 |

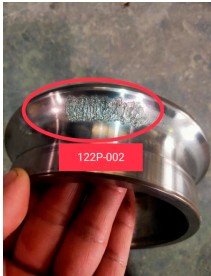

**Figure 9.** The actual faulty component of the engineering bearing.

**Table 4.** Fault characteristic frequencies.

| Type | Inner Race | Outer Race | Cage | Rolling Element |
|------|------------|------------|------|-----------------|
| 7320 | 175.38 | 120.62 | 10.11 | 98.91 |

The time domain analysis result of the selected original signal is as shown in Figure 10a based on which the transient features of the faulty bearing could be identified roughly. Apply ES analysis on the original engineering vibration signal over the full frequency band and the corresponding result is shown in Figure 10b. Unfortunately, the inner race FCF could not be extracted due to the influence of the other interferences. Apply AMF on the original signal and the de-noised signal is provided in Figure 10c, whose transient features are enhanced evidently compared with the original engineering vibration signal and the de-noising ability of AMF is proven again. Apply full frequency band ES analysis on the de-noised signal and the corresponding result is shown in Figure 10d and the enhancement effect of AMF on the impulse signal is further demonstrated because the FCF of the inner race is extracted evidently on Figure 10d. Although an inner race fault diagnosis conclusion could be obtained preliminarily based on Figure 10d, the spectral line energy components corresponding to other components are also stronger, which affects the sufficiency of the diagnosis conclusion. Section 5.1 is the same as the experimental signal; FBS analysis is required for de-noising signal using AMF to obtain better effect. The FBS analysis result of the signal shown in Figure 10c is presented in Figure 10e, based on which the key parameters (center frequency $f_c$ and bandwidth $B_w$) of the selected optimal frequency band with the biggest SI (SI = 4.2) for envelope spectral analysis are as follows: $f_c$ = 400 Hz and $B_w$ = 800 Hz. Construct an optimal band-pass filter using the obtained key parameters to further filter the de-noised signal shown in Figure 10c and the envelope of the further de-noised signal is provided in Figure 10f. At last, apply FFT on the envelope signal shown in Figure 10f and the final envelope spectral extraction result is displayed in Figure 10g, based on which the perfect effect is achieved.

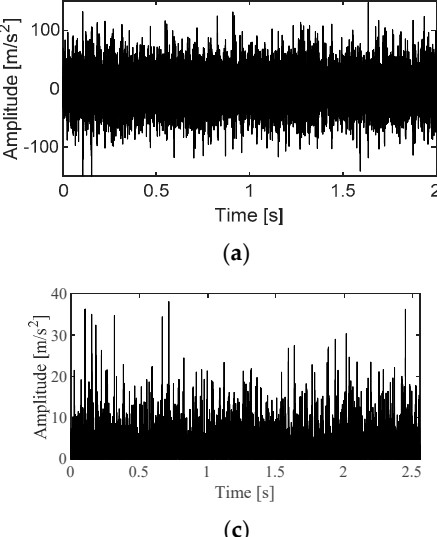

(a)

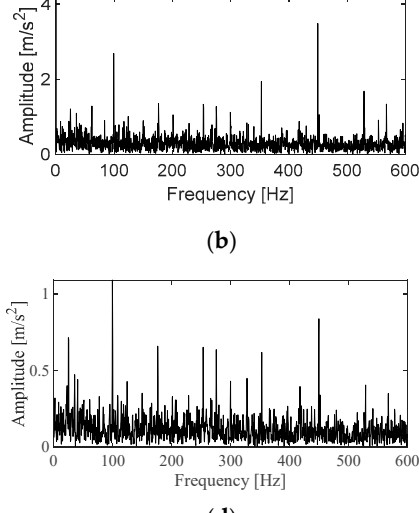

(b)

(c)

(d)

**Figure 10.** *Cont.*

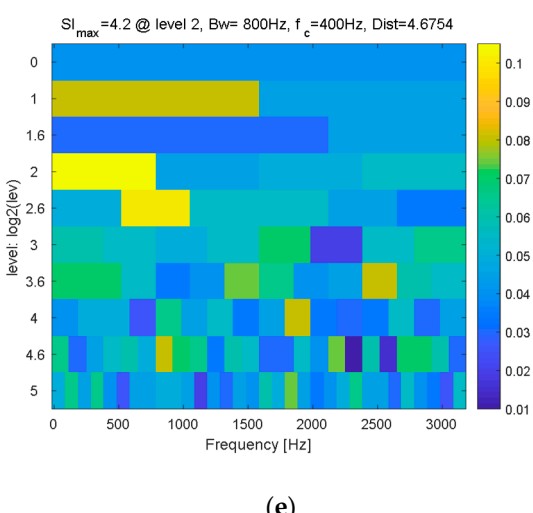

(e)

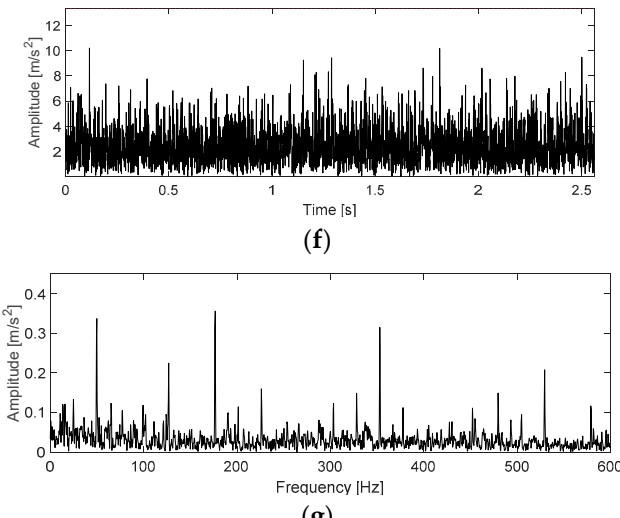

(f)

(g)

**Figure 10.** Engineering verification of the proposed method. (**a**) The original vibration signal of engineering bearing. (**b**) Envelope spectral of the signal shown in (**a**). (**c**) The filtered signal of the signal shown in (**a**) using the proposed AMF. (**d**) Envelope spectral of the signal shown in (**c**). (**e**) Analysis result of the signal shown in (**c**) using the proposed FBS method. (**f**) Envelope of the filtered signal of the signal shown in (**c**) using the proposed FBS method. (**g**) FFT of the signal shown in (**f**).

## 6. Comparison

The Mkurtogram [46] method proposed by Liao et al. is used for comparison firstly. In Mkurtgram, envelope autocorrelation analysis of the original signal is carried out for the estimation of FCF firstly. Secondly, the targeted hunting frequency zone is set automatically according to the estimated FCF. Subsequently, the analyzed signal is decomposed by 1/3 binary filter banks and the frequency domain multipoint kurtosis (FDMK) of each decomposed signal is calculated. At last, the frequency band with the maximum FDMK is selected. As verified in [46], Mkurtogram has more advantages than the classical Kurtogram method and another new FBS method, that is the frequency domain correlated kurtosis [47] method. The analyzing results of the original experimental signal using fast Mkurtogram are presented in Figure 11. Though the calculation efficiency of Mkurtogram is higher than the proposed method and the FCF of the experimental bearing's inner race is extracted as presented in Figure 11c, the harmonics of FCF are not extracted and the reason is due to the effect of the low sampling frequency when resampling being required using the Mkurtogram method. Similarly, Figure 12 provides the analyzing results of the original engineering signal using Mkurtogram, based on which not only an evident impulsive characteristic could not be observed, but also the FCF of the engineering bearing is not extracted.

The second compared method is the multi-objective informative based FBS (MOIFBS), in which the grey wolf optimizer is used for capturing the impulsiveness and cyclostationarity characteristics to adaptively determine the parameters of FBS, and its excellent performance is verified through two cases of slight bearing faults. Figure 13 presents the MOIFBS results of the original experimental signal and the impulsiveness characteristic is enhanced evidently by comparing Figure 13c with the original experimental signal. Unfortunately, the FCF of the experimental bearing's inner ring could not be identified evidently in Figure 13d. Similarly, though the FCF and its harmonics of the engineering bearing could be identified based on the analysis results as presented in Figure 13 by applying MOIFBS on the original engineering signal. By comparing Figure 13d with Figure 9, the extraction effect of the latter is better: the spectral line amplitudes located on the FCF and its harmonics are evident based on Figure 13d, but they are relatively weaker compared with the spectral line amplitude located on the rotating frequency of the experimental

bearing, which will affect the judgment of inner race fault to a certain extent. The above phenomenon does not exist in Figure 9 and the advantage of the proposed method over MOIFBS is further verified.

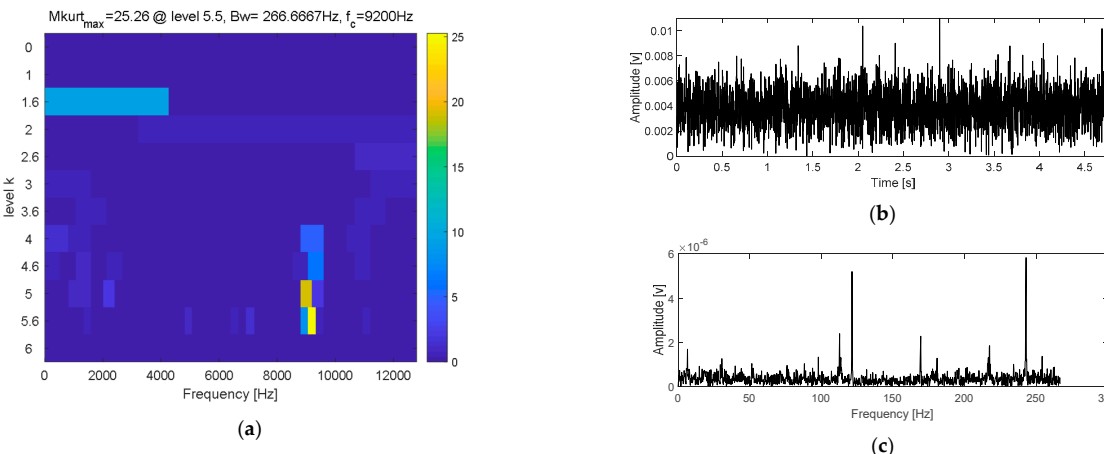

**Figure 11.** The experimental signal shown in Figure 7a using fast Mkurtogram. (**a**) FDMK. (**b**) Waveform of the filtered signal. (**c**) The envelope spectral of (**b**).

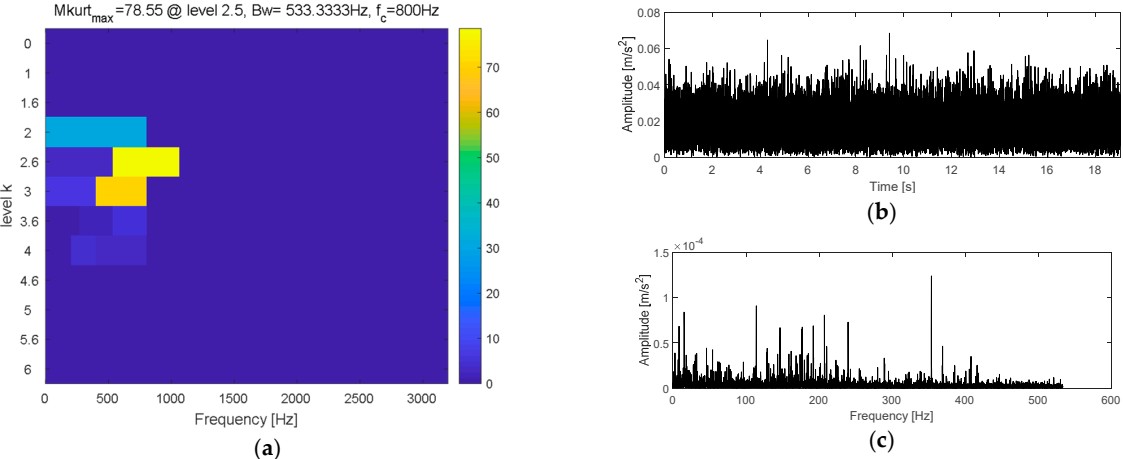

**Figure 12.** The engineering signal shown in Figure 9 using fast Mkurtogram. (**a**) FDMK. (**b**) Waveform of the filtered signal. (**c**) The envelope spectral of (**b**).

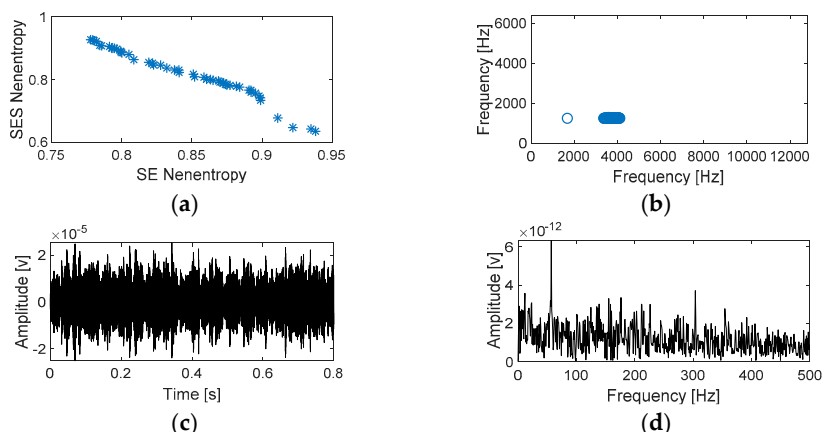

**Figure 13.** The experimental signal shown in Figure 7a using the proposed method in [48]. (**a**) Pareto front. (**b**) Distribution of the Pareto set. (**c**) Filtered signal. (**d**) The envelope spectral of (**c**).

## 7. Conclusions

A two-stage method for weak fault feature extraction of a bearing by combining an adaptive morphological filter with a new frequency band selection strategy is introduced in the paper. Firstly, a new AMF is proposed that could adaptively determine the SEs of MF according to the analyzed signal itself. The new AMF used the preprocessing method to handle the original vibration of the faulty bearing for preliminary noise reduction. Subsequently, the introduced FBS is implemented on the de-noised signal to further eliminate the influence of interferences caused by a full frequency band envelope spectral. According to the simulation verification of the new proposed AMF and FBS and the effectiveness verification of the proposed two-stage method, the following conclusions could be drawn:

1. The MHPO1 shown in Equation (13) is used as a morphological operator in the paper, which not only could de-noise the interferences preliminarily but could also enhance the transient features of the faulty bearing.
2. The proposed new AMF method could provide a new option for adaptive SEs design through simulation verification.
3. GI has the advantage of a higher reliability than the other advanced indexes such as the Hoyer measure, L2/L norm, and kurtosis index to reflect the cyclic transient features, which could be used as an index for FBS.
4. The two-stage rolling bearing weak fault feature extraction method combining an adaptive morphological filter with a new frequency band selection strategy could extract the periodic transient features of the faulty bearing much more effectively than only using one extraction method of the two proposed methods.

**Author Contributions:** Methodology, J.L.; Software, H.W.; Formal analysis, S.L.; Investigation, Q.D.; Resources, L.C. All authors have read and agreed to the published version of the manuscript.

**Funding:** The research is supported by the Key Science and Technology Research Project of the Henan Province (approved grant: 232102221039).

**Institutional Review Board Statement:** Not applicable.

**Informed Consent Statement:** Not applicable.

**Data Availability Statement:** Not applicable.

**Conflicts of Interest:** The authors declare no conflict of interest in preparing this article.

## Abbreviations

| | |
|---|---|
| AMF | Adaptive morphological filter |
| SI | Sparsity index |
| ES | Envelope spectra |
| SK | Spectral kurtosis |
| WT | Wavelet transform |
| EMD | Empirical mode decomposition |
| VMD | Variational mode decomposition |
| MF | Morphological filtering |
| GDE | Dilation and erosion gradient operator |
| GCO | Closing and opening gradient operator |
| GCOOC | Closing–opening and opening–closing gradient operator |
| AHDE | Dilation and erosion average-hat operator |
| AHCO | Closing and opening average-hat operator |
| AHCOOC | Closing–opening and opening–closing average-hat operator |
| SE | Structure element |
| FCF | Fault characteristic frequency |
| CSI | Cubic spline interpolation |

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
