# Peer review of "A Two-Stage Rolling Bearing Weak Fault Feature Extraction Method Combining Adaptive Morphological Filter with Frequency Band Selection Strategy"

_applsci, doi:10.3390/app13010668_

Round 1
Reviewer 1 Report
Summary: This paper combines morphological filtering (MF) with frequency band selection (FBS) to detect fault features of rolling bearings. The approach is applied to synthetic data and an engineering vibration signal, showing that it can extract the periodic transient features of faulty bearing effectively.
The paper heavily relies on acronyms and this makes difficult to follow it. This is aggravated by the structure of the paper: the different elements of the analysis are first presented separately in Secs. 2-3, and the whole approach is not introduced until Sec. 4. The paper could be improved by explaining first the methodology/objectives and then presenting the different components.
In general, the clarity of the exposition could be improved by limiting acronyms and unnecessary definitions (see Sec. 2 for instance).
Suggestions:
1.- Some letters denoting sub-figures are missing in Figs. 2, 6 and 8.
2.- What is exactly depicted in Figs. 6(e) and 8(e)? Where is level defined?
Author Response
Dear reviewer:
I have revised the paper thoroughly under your comments. Thanks a lot for your valuable advice.
hongchao wang
12.9.2022

Reviewer 2 Report
The paper reports an interesting and useful mathematical analysis along with experimental work, well structured in the manuscript. The manuscript has some weaknesses. Mentioned below aspects should be taken into consideration during the revision:
I suggest adding "Nomenclature" section in the manuscript as a large number of abbreviations have been used in the manuscript.
I suggest add a brief explanation of the experiment.
In Fig. 2, 6 & 8 please check the caption of the sub-images in the figure
Author Response

(The authors gave the same response as above.)
